# Association of Vitamin D Supplementation with Glutathione Peroxidase (GPx) Activity, Interleukine-6 (IL-6) Levels, and Anxiety and Depression Scores in Patients with Post-COVID-19 Condition

**DOI:** 10.3390/ijms26104582

**Published:** 2025-05-10

**Authors:** Iván Ignacio-Mejía, Cindy Bandala, José Francisco González-Zamora, Leslie Chavez-Galan, Ivette Buendia-Roldan, Karina Pérez-Torres, María Zobeida Rodríguez-Díaz, Denilson Xipe Pacheco-Tobón, Saray Quintero-Fabián, Marco Antonio Vargas-Hernández, Humberto Carrasco-Vargas, Ramcés Falfán-Valencia, Gloria Pérez-Rubio, Kevin Alexis Hernández-Lara, Saúl Gómez-Manzo, Daniel Ortega-Cuellar, Fredy Ignacio-Mejía, Noemí Cárdenas-Rodríguez

**Affiliations:** 1Laboratorio de Medicina Traslacional, Escuela Militar de Graduados de Sanidad, Universidad del Ejército y Fuerza Aérea, Mexico City 11200, Mexico; ivanignacio402@gmail.com (I.I.-M.); yaras_44@yahoo.com.mx (S.Q.-F.); 2Escuela Superior de Medicina, Instituto Politécnico Nacional, Mexico City 11340, Mexico; crodriguezba@ipn.mx; 3Centro de Investigación Traslacional, Instituto Nacional de Pediatría, Secretaría de Salud, Mexico City 04530, Mexico; jf.gonzalezzamora@gmail.com; 4Laboratorio de Inmunología Integrativa, Instituto Nacional de Enfermedades Respiratorias Ismael Cosio Villegas, Mexico City 14080, Mexico; lchavez_galan@iner.gob.mx; 5Laboratorio de Investigación Traslacional en Envejecimiento y Enfermedades Fibróticas, Instituto Nacional de Enfermedades Respiratorias Ismael Cosio Villegas, Mexico City 14080, Mexico; ivettebu@yahoo.com.mx (I.B.-R.); mayode1997@hotmail.com (K.P.-T.); zobeidadz3@gmail.com (M.Z.R.-D.); xipept@icloud.com (D.X.P.-T.); 6Subdirección de Investigación, Escuela Militar de Graduados de Sanidad, Universidad del Ejército y Fuerza Aérea, Mexico City 11200, Mexico; mavh78@yahoo.com.mx; 7Dirección de la Escuela Militar de Medicina, Universidad del Ejército y Fuerza Aérea, Mexico City 11200, Mexico; hcarrascovargas@gmail.com; 8Laboratorio de HLA, Instituto Nacional de Enfermedades Respiratorias Ismael Cosio Villegas, Mexico City 14080, Mexico; rfalfanv@iner.gob.mx (R.F.-V.);; 9Laboratorio de Neurociencias, Instituto Nacional de Pediatría, Secretaría de Salud, Mexico City 04530, Mexico; itskevinlara@outlook.com; 10Laboratorio de Bioquímica Genética, Instituto Nacional de Pediatría, Secretaría de Salud, Mexico City 04530, Mexico; saulmanzo@ciencias.unam.mx; 11Laboratorio de Nutrición Experimental, Instituto Nacional de Pediatría, Secretaría de Salud, Mexico City 04530, Mexico; dortegadan@gmail.com; 12Hospital Central Militar, Mexico City 11200, Mexico; frime858@gmail.com

**Keywords:** post-COVID-19 condition, long-COVID-19, vitamin D_3_ supplementation, cholecalciferol, interleukin-6, glutathione peroxidase activity, respiratory symptoms, anxiety, depression

## Abstract

Coronavirus disease 2019 (COVID-19) presents with various symptoms, and some patients develop post-COVID-19 condition (PCC). Vitamin D has shown therapeutic potential in COVID-19 and may offer benefits for PCC. The aim of this study was to evaluate the differences associated with two supplementation strategies (bolus and daily) on interleukin-6 (IL-6) levels, glutathione peroxidase (GPx) activity, and clinical outcomes in PCC patients, regardless of whether target 25 (OH) D levels reached the ideal range. We conducted a self-controlled study in which 54 participants with PCC were supplemented with vitamin D_3_ (n = 28 bolus and n = 26 daily) for 2 months. Blood samples were collected to measure IL-6 levels and GPx activity using spectrophotometric methods. The Hospital Anxiety and Depression Scale (HADS) was used to assess mental function. Both bolus and daily vitamin D supplementation were significantly associated with increased GPx activity and decreased IL-6 levels. Daily supplementation was additionally associated with a significant reduction in anxiety and depression scores. However, neither regimen was associated with improvements in cough, dyspnea, or fatigue. These findings suggest a potential association between vitamin D supplementation and improvements in antioxidant and neuropsychiatric parameters in PCC, possibly mediated by its immunomodulatory and antioxidant properties. Further placebo-controlled trials are warranted to determine whether these observed associations reflect causal relationships.

## 1. Introduction

In March 2020, the World Health Organization (WHO) declared COVID-19 a global pandemic. Since then, COVID-19 has caused a significant global health crisis, with more than 774 million confirmed cases and 7.05 million reported deaths by 2024 [1]. SARS-CoV-2 infection can cause a broad spectrum of clinical presentations, ranging from mild to severe forms, including multi-organ dysfunction [2]. After the resolution of the acute infection, some patients continue to experience persistent symptoms, including cardiorespiratory manifestations (chest pain, dyspnea, cough, sputum production), neurological impairments (cognitive dysfunction, memory loss, headaches), fatigue, and psychiatric disturbances such as depression and anxiety. When persistent, these symptoms may evolve into post-COVID-19 condition (PCC) [3,4,5], which is defined as the persistence of one or more symptoms for more than two to three months after the onset of COVID-19, with no alternative medical explanation and generally have an impact on everyday functioning. Symptoms might be new onset after initial recovery from an acute COVID-19 episode or persist from the initial illness [6,7,8]. Importantly, PCC symptoms may fluctuate over time, with some symptoms reappearing or varying in intensity during follow-up, which adds complexity to their evaluation and monitoring [8]. PCC has been reported in up to 87% of COVID-19 survivors, with at least 65 million cases documented by 2023 [9,10].

The pathophysiology of PCC remains incompletely understood. A leading hypothesis is that persistent SARS-CoV-2 infection triggers chronic inflammation, leading to prolonged symptoms [11,12,13]. An additional proposed mechanism involves immune dysregulation, as studies have reported T-cell alterations in PCC patients, suggesting a persistent inflammatory response [10,14,15,16,17]. Elevated levels of pro-inflammatory cytokines such as IL-6, IL-8, IL-10, IL-18, interferon-γ (IFN-γ), monocyte chemoattractant protein-1 (MCP-1), interferon-γ-inducible protein-10 (IP-10), and tumor necrosis factor (TNF) have been reported in affected individuals [2,17,18,19,20,21,22]. These inflammatory changes are accompanied by increased activation of NADPH oxidase (NOX) activation and excessive ROS production, contributing to prolonged tissue damage. At the same time, antioxidant defenses are compromised, as evidenced by reduced superoxide dismutase (SOD) activity and decreased glutathione (GSH) levels in serum, plasma, lymphocytes, and monocytes [2,17,18,19,20,21,22]. Among antioxidant enzymes, glutathione peroxidase (GPx) plays a critical role in neutralizing oxidative damage. However, limited studies have evaluated its activity in COVID-19 patients. One study reported significantly lower GPx activity in erythrocytes of COVID-19 patients compared to healthy controls [23]. Additionally, patients with severe COVID-19 and acute respiratory distress syndrome (ARDS) admitted to intensive care units (ICUs) exhibited reduced serum GPx3 activity, which was significantly improved after 10–14 days of selenium supplementation [24]. Emerging genetic evidence suggests that a specific GPx3 variant (rs8177412) may be associated with an increased risk of developing severe COVID-19 and cardiovascular complications in recurrent cases [25,26]. These findings support the role of GPx activity as a potential antioxidant marker in PCC, reinforcing the link between oxidative stress and persistent symptoms [23,24,25,26,27]. Moreover, vitamin D may modulate interleukin-6 (IL-6) levels through its immunomodulatory effects [27].

Clinical studies have associated vitamin D deficiency with increased mortality among hospitalized COVID-19 patients and have linked vitamin D supplementation to decreased disease severity [28,29,30,31,32]. In the context of PCC, recent studies have reported a high prevalence of vitamin D insufficiency among affected individuals compared to the general population, suggesting a possible association with delayed recovery [33,34,35]. Although vitamin D supplementation has been proposed as a potential therapeutic option for alleviating PCC symptoms, published findings have been inconsistent, and the underlying mechanisms remain unclear [36,37,38,39,40].

Vitamin D, in its active form 1,25-(OH)_2_D, acts as a steroid hormone regulating multiple cellular functions through the nuclear vitamin D receptor (VDR), which is expressed in a wide range of tissues [27,41]. Its actions involve both genomic effects, by modulating the transcription of genes related to inflammation, antioxidant responses, and calcium homeostasis [42], and non-genomic effects, including the activation of intracellular pathways associated with cell survival and tissue repair [27,43,44]. The synthesis and bioavailability of the active form of vitamin D depend directly on the serum concentrations of its precursor, 25(OH)D, which is synthesized in the liver from vitamin D_3_ and subsequently primarily converted in the kidneys, but also in peripheral tissues such as neurons, glial cells, and immune cells, through the enzyme CYP27B1 [27,45,46,47,48].

Emerging evidence from pharmacokinetic studies highlights that not only the absolute concentration of 25(OH)D but also the rate of increase may influence biological responses [30,49]. Following vitamin D_3_ supplementation, the serum response of 25(OH)D is not immediate; it requires several weeks for vitamin D_3_ to be absorbed, stored (especially in adipose tissue in obese individuals), and metabolized [49]. The kinetic response follows an exponential curve, generally reaching a steady state after several weeks of supplementation. However, transient increases in 25(OH)D concentrations following bolus supplementation may induce early biological effects before steady-state levels are achieved. The magnitude of 25(OH)D increase is proportional to the administered daily dose and body weight, with approximately 2.5 IU/kg/day of vitamin D_3_ required to raise serum 25(OH)D levels by 1 ng/mL [49].

In this context, the regimen of administration can significantly influence the pharmacokinetics of vitamin D and, consequently, its clinical effects. Faster increases in serum 25(OH)D levels achieved through bolus regimens may differentially affect oxidative stress and inflammatory pathways compared to the gradual increases obtained through daily supplementation [37]. This difference in the kinetic response is explained by the fact that, although both daily and bolus supplementation regimens are effective in raising serum 25(OH)D levels, bolus administration results in a significantly faster increase compared to lower daily doses [38,39,40,50,51].

Furthermore, individual variability in the response to vitamin D_3_ supplementation has been partially attributed to genetic factors, including the activity of enzymes such as CYP2R1, CYP24A1, and the synthesis of vitamin D-binding proteins [49,52,53,54,55]. Some studies have suggested that daily supplementation may be more effective than intermittent bolus dosing in preventing COVID-19 [56], while other findings indicate that a combined strategy, an initial bolus followed by daily doses, may be more efficient in increasing 25(OH)D concentrations compared to daily dosing alone [57]. Observational studies have suggested that 25(OH)D concentrations above 38 ng/mL may be associated with improvements in specific inflammatory biomarkers among COVID-19 populations [58].

While some studies suggest that daily supplementation may be more effective for neuropsychiatric symptoms such as anxiety and depression, bolus administration has been linked to greater reductions in inflammatory markers, particularly IL-6 [59,60].

To date, no interventional study has specifically evaluated the association of vitamin D_3_ supplementation with IL-6 levels, GPx activity, and clinical outcomes in patients with PCC, either within the Mexican population or globally. We hypothesize that vitamin D_3_ supplementation is associated with modifications in GPx activity, IL-6 levels, and clinical symptoms in patients with PCC, independent of changes in serum 25(OH)D concentrations. This study aims to evaluate whether vitamin D_3_ supplementation is associated with GPx activity, IL-6 levels, and clinical outcomes in patients with PCC, comparing the impact of bolus versus daily supplementation strategies.

## 2. Results

### 2.1. Clinical and Demographic Data

A total of 54 participants were enrolled, including 28 in the bolus supplementation group and 26 in the daily supplementation group. Table 1 summarizes the age, sex, body mass index (BMI), and clinical parameters of the treatment groups. Baseline demographic and clinical characteristics were comparable between groups.

Vitamin D levels were quantified before and after cholecalciferol supplementation in both treatment groups. Before supplementation, 39.3% of patients in the bolus regimen group exhibited insufficient 25(OH)D levels (21–29.9 ng/mL), while 60.7% had deficient levels (<20 ng/mL). Among patients in the daily regimen group, 3.8% had adequate 25(OH)D levels (30–39.9 ng/mL), 30.8% had insufficient levels, and 65.4% were deficient.

Changes in 25(OH)D levels were evident after vitamin D supplementation. In the bolus regimen group, 4.3% of patients remained below sufficiency (<30 ng/mL), while 52.2% achieved 25(OH)D levels between 30 and 39.9 ng/mL, and 43.5% reached levels between 40 and 60 ng/mL In the daily supplementation group, 27.3% remained below sufficiency, while 36.4% achieved 30–39.9 ng/mL, and 36.4% reached 40–60 ng/mL In the daily supplementation group, 27.3% of patients remained below the sufficiency threshold (<30 ng/mL), 36.4% achieved 25(OH)D levels between 30 and 39.9 ng/mL, and 36.4% reached higher levels ranging from 40 to 60 ng/mL Table 2 shows serum 25(OH)D levels, cholecalciferol requirements, and daily doses required for both treatment groups. Baseline 25(OH)D levels were comparable between the groups (bolus: 19.20 ± 5.59 ng/mL, daily: 18.99 ± 5.04 ng/mL; *p* = 0.88). The required increase in 25(OH)D levels was comparable between groups (bolus: 30.94 ± 5.64 ng/mL, daily: 31.00 ± 5.04 ng/mL; *p* = 0.87). Significant differences were observed in final 25(OH)D levels (bolus: 42.64 ± 10.38 ng/mL, daily: 35.99 ± 8.85 ng/mL; *p* = 0.024) and in the changes in 25(OH)D levels (bolus: 23.99 ± 11.65 ng/mL, daily: 17.28 ± 9.28 ng/mL; *p* = 0.037). Although the mean calculated daily cholecalciferol dose was higher in the bolus group (6263.79 ± 1627 IU vs. 5796.36 ± 1437 IU), this difference was not statistically significant (*p* = 0.27).

### 2.2. Follow-Up and Outcomes

#### 2.2.1. Principal Outcomes

Table 3 shows the serum measurements of GPx activity and IL-6 levels pre-treatment and post-treatment (self-controlled analysis). GPx activity increased in all patients group (*p* = 0.0001) and in both the bolus (*p* = 0.006) and daily (*p* = 0.01) supplementation groups. A significant decrease in serum IL-6 concentration was observed only in the bolus group (*p* = 0.01), with no significant change in all patients (*p* = 0.07) and in daily group analysis (*p* = 0.30).

In the comparative analysis (bolus vs. daily), no significant differences were observed in post-treatment measurements and for IL-6 pre-treatment levels (*p* > 0.05) but if for GPx activity during pre-treatment (*p* = 0.006) (Appendix A).

#### 2.2.2. Secondary Outcomes

Table 4 shows the results of the Hospital Anxiety and Depression Scale pre-treatment and post-treatment (self-controlled analysis). Anxiety and depression scores decreased significantly in all patients (*p* = 0.0001 and *p* = 0.001 for anxiety and depression, respectively). Patients receiving bolus supplements did not show significant differences (*p* > 0.05) in either parameter, while patients receiving daily supplements showed significant decreases in anxiety and depression scores (*p* = 0.0001 for both).

In the comparative analysis (bolus vs. daily) a significant difference was observed only in the post-treatment evaluation of the HADS anxiety score (*p* = 0.01) (Appendix A).

Post-COVID-19 symptoms were evaluated in those participants who reported them at baseline, specifically cough, dyspnea, and fatigue, which were the most prevalent symptoms. A reduction was observed in cough (29.41%, 5/17), dyspnea (57.14%, 8/14), and fatigue (58.33%, 7/12) following cholecalciferol supplementation (see Table 5). When we analyzed by treatment group, the bolus supplementation group showed improvements of 33.3% (3/9) in cough, 50% (4/8) in dyspnea, and 80% (4/5) in fatigue. Similarly, the daily dose group showed improvements of 50% (4/8) in cough, 66.6% (4/6) in dyspnea, and 42.85% (3/7) in fatigue; however, these improvements were not statistically significant (*p* > 0.05).

#### 2.2.3. Safety Outcomes

The maximum dose administered did not exceed 50,000 IU per day, and no patient achieved serum vitamin D (25(OH)D) levels above 60 ng/mL. In addition, potential adverse effects were continuously monitored throughout this study. Particular attention was given to potential adverse effects associated with vitamin D administration, such as hypercalcemia, muscle weakness, fatigue, headache, dry mouth, nausea, and dizziness, which are indicative of excessive vitamin D levels. No signs of toxicity or adverse effects were reported during the study period.

## 3. Discussion

Oxidative stress and chronic inflammation are important factors in the pathogenesis of PCC [61,62,63,64,65,66,67,68,69,70,71,72]. Vitamin D has been suggested as a potential therapeutic agent due to its immunomodulatory and antioxidant properties [27]. In this study, vitamin D was administered exclusively by the oral route, either as a daily dose or as a weekly bolus. This route was selected for ethical and practical reasons, ensuring safety, ease of use, and alignment with outpatient treatment standards commonly applied in PCC populations. Our results showed that vitamin D supplementation was associated with an increased GPx activity in both regimens, significantly reduced IL-6 levels in the bolus group, and improvements in anxiety and depression scores, particularly among patients receiving daily supplementation. These findings are consistent with previous research suggesting that vitamin D may regulate oxidative stress, inflammation, and neuropsychiatric symptoms, with pharmacokinetic variations potentially influencing these effects [27,34,37].

Unlike most previous studies, our work evaluates the effect of vitamin D supplementation on IL-6 and GPx activity, using two distinct oral regimens in a Mexican cohort with PCC. To our knowledge, this study provides a novel contribution to understanding the antioxidant and immunological roles of vitamin D in PCC. In particular, the differences observed between bolus and daily regimens underscore the distinct physiological effects each may exert. Most patients had vitamin D deficiency (<20 ng/mL), possibly related to glutathione (GSH) depletion [35,73,74]. More than half of the participants were overweight or obese, which can impair vitamin D metabolism and potentially increase susceptibility to PCC [75,76]. One-third of the participants had diabetes or hypertension, conditions that may affect the immunomodulatory effects of vitamin D [77,78,79,80].

GPx is essential for maintaining redox homeostasis and cellular defense [81,82]. Both supplementation regimens were associated with increased GPx activity, supporting the potential antioxidant function of vitamin D. This is consistent with studies showing that vitamin D regulates glutathione metabolism and activates the Nrf2 pathway [83,84,85,86]. PCC is associated with reduced GPx activity and increased oxidative stress, suggesting an imbalance in redox homeostasis [87]. Our findings indicate a potential association between vitamin D supplementation and increased GPx activity, possibly through glutathione synthesis and Nrf2-mediated antioxidant responses [88,89].

SARS-CoV-2 may impair GPx function through its main protease (M^pro^), contributing to elevated oxidative stress. Studies of M^pro^-transfected cells have shown increased ROS production, further supporting the role of GPx in mitigating viral-induced oxidative damage [83,84,85,86]. Vitamin D supplementation was associated with increased GPx activity in both regimens, suggesting a potential link to enhanced antioxidant defenses in PCC, which may help mitigate mitochondrial dysfunction and endothelial damage [61,62,63,64,65,66,67,68,69,70,71,72].

IL-6 is a key inflammatory mediator in both PCC and severe COVID-19 [65,90,91,92,93]. Persistent elevation of IL-6 has been linked to greater disability and neurological complications in PCC patients, although findings across studies remain inconsistent [94,95,96,97,98,99,100]. Our findings revealed a significant reduction in IL-6 levels among participants in the bolus group, whereas no such change was observed in the daily supplementation group, suggesting that the rapid elevation of serum vitamin D levels induced by bolus supplementation may promote stronger anti-inflammatory responses. Vitamin D modulates immune responses through multiple pathways, including downregulation of Toll-like receptors (TLRs), suppression of C-reactive protein (CRP), and modulation of cytokine production [101,102,103,104], and supplementation has been shown to lower IL-6 levels, particularly in severe cases where its deficiency is associated with cytokine storm [105,106,107,108]. In COVID-19 survivors, chronic inflammation and immune dysregulation have been associated with factors including biological aging, insulin resistance, metabolic disorders, reduced sun exposure, physical inactivity, and smoking [109,110]. Additionally, sex, obesity, ethnicity, and pre-existing inflammatory conditions have been identified as risk factors for elevated IL-6 levels in COVID-19 patients [104,108]. Persistent IL-6 elevation in PCC has been observed in patients with comorbidities such as diabetes and hypertension, contributing to a chronic low-grade inflammatory state [111,112]. Based on these findings, vitamin D supplementation, especially in a bolus regimen, appears to be associated with a reduction in inflammatory markers in PCC patients; however, further research is needed to establish its clinical relevance [38,110].

Studies suggest that bolus vitamin D supplementation can rapidly reduce IL-6, IL-8, and TNF levels, suggesting that higher initial serum concentrations may have a greater anti-inflammatory effect [60]. The greater reduction in IL levels observed in our bolus group may be related to the rapid achievement of higher vitamin D levels. Although both regimens effectively increased serum 25(OH)D concentrations, their distinct pharmacokinetic profiles may lead to differential immunomodulatory effects over time. Future research should explore whether sustained daily dosing over a longer period of time could achieve similar anti-inflammatory effects, particularly in PCC patients with chronic inflammation.

Recent studies have associated low vitamin D levels, anxiety, and depression with PCC, suggesting their potential role as prognostic factors for predicting long-COVID in the outpatient setting [113]. In addition, anxiety and depression are more common in PCC patients with gut–brain interaction disorders [114]. Vitamin D supplementation was significantly associated with reductions in anxiety and cognitive symptoms in patients with PCC. These findings are consistent with previous studies suggesting its role in modulating neuroinflammation [11,113,115]. Vitamin D has been associated with improved neurological health outcomes [116,117,118]. One clinical trial showed that weekly supplementation with 60,000 IU for eight weeks reduced anxiety and cognitive symptoms in patients with PCC [119]. Another study found an association between hypomagnesemia, vitamin D deficiency, and increased anxiety and depression in long-COVID-19 patients [120]. Given its neuroprotective role, vitamin D has been proposed as a potential therapeutic agent for the neuropsychiatric manifestations of long-COVID syndrome [37]. GPx has also been proposed to function as a neuromodulator in neuropsychiatric conditions due to its neuroprotective role against oxidative stress and its involvement in the Nrf2-mediated glutathione synthesis pathway [121]. Another study found that the impact of acute COVID-19 on affective symptoms in long-COVID (anxiety and depression) were partially mediated by reduced GPx activity, lower zinc levels, and increased myeloperoxidase and nitric oxide production [87]. In our study, the observed increase in GPx activity following vitamin D supplementation may be linked to the reduction in neuropsychiatric symptoms in PCC patients [122]. The variability in response to vitamin D supplementation between the bolus and daily dosing groups suggests that the dosing regimen may influence its biological effects. In this study, the daily regimen was associated with reductions in anxiety and depression symptoms, whereas the bolus regimen was associated with a greater reduction in IL-6 levels.

These observed differences may stem from the distinct pharmacokinetic profiles of the two supplementation strategies. Daily dosing maintains more stable serum vitamin D levels, which may be associated with sustained neuroprotective effects and modulation of neuropsychiatric symptoms over time [123,124]. In contrast, bolus administration produces transiently higher peaks in serum 25(OH)D concentrations, which may be associated with a more immediate immunomodulatory response, potentially explaining its greater impact on IL-6 reduction [60].

Pharmacokinetic differences between vitamin D supplementation strategies have been well documented, particularly with regard to dose-dependent serum responses. Higher oral doses have been associated with a faster and more pronounced increases in serum 25(OH)D concentrations [51,60]. Such pharmacokinetic variations may modulate the biological activity of vitamin D, including its antioxidant and immunomodulatory effects [30,32,60]. Based on this rationale, our study compared daily and bolus supplementation regimens to explore their potential differential impact on clinical and biochemical outcomes in patients with PCC.

Previous studies comparing both dosing regimens in other clinical settings suggest that fractionated supplementation promotes a more regulated metabolism of vitamin D and reduces the catabolic activation of CYP24A1 [123]. However, other research suggests that bolus administration is associated with rapid reductions in inflammatory cytokines such as IL-6, IL-8, and TNF-α, suggesting an immediate immunomodulatory effect [60]. Our findings support the notion that dosing regimen may play a role in modulating the effects of vitamin D supplementation in PCC. Further studies should investigate whether prolonged daily supplementation can achieve anti-inflammatory effects comparable to the bolus regimen. No significant differences in overall clinical outcomes were observed between the two supplementation regimens in terms of IL-6 levels, but daily supplementation was significantly associated with decreased anxiety and depression scores compared to the bolus group. Daily supplementation has been reported to be more effective than bolus dosing in increasing 25(OH)D levels because it maintains a stable 24,25(OH)_2_D_3_/25(OH)D_3_ ratio over 28 days. In contrast, bolus supplementation results in a transient increase in the 24,25(OH)_2_D_3_/25(OH)D_3_ ratio within seven days, suggesting a sustained but possibly prolonged catabolic activation of CYP24A1 [123,124]. These findings suggest that fractionated administration of vitamin D may be associated with more stable serum levels, which may be relevant for its potential long-term effects, including neurological symptoms [59]. Nevertheless, the differential effects between bolus and daily supplementation suggest that future studies should explore whether sustained daily dosing could achieve similar anti-inflammatory benefits. Vitamin D supplementation was linked to modest, though statistically non-significant, reductions in respiratory symptoms, including cough, dyspnea, and fatigue. However, these improvements did not reach statistical significance, suggesting that vitamin D may have a limited effect on resolution of respiratory symptoms in PCC. It is important to note that due to the fluctuating course of PCC symptoms, only participants with specific symptoms at baseline were followed throughout the intervention. This strategy was intended to reduce variability and improve the interpretability of changes.

This aligns with previous studies reporting no significant association between baseline vitamin D status and persistent respiratory symptoms following COVID-19, nor substantial improvements after supplementation [33,125,126,127,128]. Although vitamin D may support overall recovery, its effectiveness in alleviating persistent respiratory symptoms in PCC remains uncertain. Further research is needed to determine whether specific subgroups of PCC patients may benefit from vitamin D intervention.

This study has several strengths and limitations. The randomized and partially blinded design strengthened the internal validity of this study. Participants, outcome assessors, and the data analyst remained blinded to group allocation throughout this study. Although the personnel responsible for administering the supplementation were not blinded, their role was limited exclusively to the allocation stage and they had no further involvement in data collection, outcome evaluation, or analysis. In-person monitoring further enhanced adherence and ensured high-quality data collection. However, because most participants had insufficient or deficient vitamin D levels, supplementation was ethically justified. This may limit the generalizability of our findings to individuals with adequate baseline vitamin D levels.

Furthermore, our findings should be interpreted with caution because of other limitations to generalizability. This study was conducted at a single center in Mexico, most participants had comorbidities such as obesity, type 2 diabetes, or hypertension, conditions that may influence both vitamin D metabolism and treatment response. In particular, several studies have suggested an association between vitamin D status and cardiovascular risk and disease (hypertension, diabetes, obesity, coronary artery calcification, stroke and heart disease). Moreover, the effect of vitamin D on IL-17 and IL-6 levels remain controversial due to the variability in the treatment duration, route, and frequency of administration, as well as the dosage of vitamin D supplementation. Other factors such as age, sex, BMI and smoking may complicate the association between vitamin D levels and inflammatory/antioxidant status and mental health. In addition, all patients had a history of severe COVID-19 requiring hospitalization, which may not be representative of the broader PCC population, particularly those with milder disease who did not require intensive care. Therefore, multicenter studies with more diverse populations are needed to validate and extend these findings.

A self-controlled design was selected given the exploratory nature of this study and the early stage of vitamin D research in PCC. This design is especially useful for evaluating biochemical changes within a short intervention period and for controlling confounding variables that remain stable over time [129,130,131]. However, we acknowledge that self-controlled studies may overestimate effect sizes compared to controlled trials and may be subject to residual bias. As such, this study cannot establish causality, only associations [129]. Finally, this study, being self-controlled, does not establish a cause-effect relationship, but only an association of facts, without indicating causality. The advantage of this design is that it allows for a simple investigation while maintaining homogeneity of confounding variables across groups.

Our findings highlight the potential role of vitamin D in the PCC management. Given its immunomodulatory and antioxidant properties, early detection and correction of vitamin D deficiency should be prioritized in COVID-19 survivors. Moreover, we observed improvements in clinical symptoms following supplementation, which may have positively impacted patients’ well-being. The treatment was administered in a safe and simple manner, with no adverse effects reported, and was well tolerated. At follow-up, no recurrence of neuropsychiatric symptoms was reported, suggesting potential persistence of the observed benefits.

Future research should evaluate the long-term effects of vitamin D supplementation in different PCC subgroups, taking into account the potential benefits of personalized dosing strategies.

Given the global burden of PCC and its socioeconomic impact, vitamin D supplementation could represent a cost-effective and widely accessible strategy to mitigate the long-term effects of COVID-19. These findings contribute to the broader understanding of post-viral syndromes and may inform therapeutic approaches beyond PCC, particularly in conditions characterized by chronic inflammation and oxidative stress.

## 4. Materials and Methods

### 4.1. Trial Oversight

The protocol was approved by the Ethics Committee of the National Institute of Respiratory Diseases Ismael Cosío Villegas (INER) under protocol number C32-22. Written informed consent was obtained from all participants. All procedures adhered to the ethical standards of the 1964 Declaration of Helsinki and the regulations of the Institutional Ethics Committees. The study design and data are publicly available in the ClinicalTrials.gov registry under identifier NCT06419712.

### 4.2. Patients

This was a prospective, interventional self-controlled study conducted in patients diagnosed with post-COVID according to the World Health Organization (WHO) definition [132]. Patients were recruited from a post-COVID cohort at the National Institute of Respiratory Diseases (INER) and underwent direct clinical evaluation at the time of study enrollment. All patients were over 18 years of age and had completed at least 12 months of follow-up after being discharged from hospital for a severe COVID-19 episode and subsequently meeting WHO diagnostic criteria for PCC. At enrollment, patients were assessed for persistent post-COVID symptoms, including dyspnea, fatigue, cough, depression, or anxiety. Serum 25-hydroxyvitamin D (25(OH)D) levels were measured immediately before enrollment using a chemiluminescent immunoassay, confirming vitamin D deficiency (<30 ng/mL) in all participants as an eligibility criterion. Although serum 25(OH)D levels were measured prior to randomization, they were not used as a stratification factor for group allocation. Randomization was conducted independently of baseline 25(OH)D concentrations, which were used afterward solely to calculate individualized supplementation doses. Patients were recruited between December 2022 and March 2023. Patients were excluded if they met any of the following criteria: difficulty reading or writing, inability to perform functional walking or spirometry tests, recent vitamin D supplementation, history of neurological or psychiatric disorders, substance abuse, use of psychiatric medications, cancer, rheumatologic diseases, advanced chronic diseases, acute cardiovascular events, muscular disorders, untreated non-respiratory chronic diseases (e.g., diabetes mellitus, systemic arterial hypertension, hypothyroidism, epilepsy), enrollment in other clinical trials, pregnancy, or breastfeeding. Retrospective data were not collected. All clinical and demographic information was obtained through structured interviews and direct clinical assessments at enrollment.

Given that all participants had serum 25(OH)D levels classified as insufficient or deficient at the time of enrollment, randomization to an un-supplemented control group was considered ethically inappropriate. The trial did not include a placebo group as this was not approved by the ethics committee. The patients had symptoms of prolonged COVID-19, and the physicians on the committee did not believe it was ethical to deny them a treatment option that could alleviate their symptoms in the context of a nutritional deficiency that could pose unnecessary risks. The rationale for this decision is mentioned in the ethical guidelines that it is the responsibility of the researcher to act in the best interest and benefit and ensure the health of patients, especially when there is the possibility of a preventive, adequate and existing therapeutic [133,134,135]. Accordingly, the ethical dilemma was to have a group of patients who did not receive an existing intervention whose benefit was beyond doubt, in this case vitamin D supplementation, and in addition the disease (post-COVID-19 syndrome) resulted in significant physical and neuropsychiatric manifestations [136,137,138].

### 4.3. Randomization

Randomization was performed at study baseline by an independent investigator using a computer-generated list with a block size of four. Randomization was performed without taking into account individual baseline 25(OH)D concentrations. Patients were randomly assigned to one of two groups: (a) 26 patients in the daily supplementation group and (b) 28 patients in the bolus supplementation group. The randomization process followed a 1:1 allocation ratio. Group assignment was concealed from participants and outcome assessors throughout the study period (see Appendix A).

### 4.4. Intervention

Vitamin D supplementation was administered exclusively by the oral route in an outpatient setting due to the fact that all participants were not hospitalized and because of its well-established safety profile, ease of administration, and practicality in routine clinical settings. This strategy is consistent with current clinical guidelines for the management of vitamin D deficiency in ambulatory patients and allows for individualized dose titration without the need for invasive procedures.

All patients received oral vitamin D supplementation in one of two dosing regimens: a daily administration or a weekly oral bolus (both using 4000 IU cholecalciferol dispersible tablets, Histofil, Medix, Mexico City, Mexico). This study evaluated the effects of vitamin D supplementation without using serum 25(OH)D levels as the primary outcome measure. At baseline and at the beginning of the second month, each patient’s cholecalciferol dose was adjusted based on their baseline serum 25(OH)D concentration to reach the target level of 50 ng/mL [116,139]. The required daily dose (IU) was determined using the following formula: Daily dose (IU) = [body weight (kg) × desired increase in 25(OH)D (ng/mL) × 2.5] − 10 [49]. In the bolus regimen, the weekly dose was calculated as seven times the determined daily dose and was administered once per week for eight weeks.

Clinical evidence suggests that higher oral doses result in faster and more pronounced increases in serum 25(OH)D levels [51,52,53,54,55,56,57,58,59]. These metabolic differences may influence the biological effects of vitamin D [30,32,60]. Therefore, this study compared daily and bolus supplementation strategies to explore their potential differential impact on clinical and biochemical outcomes in patients with PCC.

### 4.5. Blinding

Throughout this study, outcome assessors, the statistician responsible for data analysis, and the participants remained blinded to group assignment. Participants were unaware of the existence of an alternate study group and had no contact with other participants, reinforcing the strict blinding procedures and ensuring the integrity of this study.

### 4.6. Outcomes

The primary outcomes included glutathione peroxidase (GPx) activity and serum interleukin-6 (IL-6) levels, assessed at baseline and day 60.

Secondary outcomes included neuropsychiatric symptoms (anxiety and depression) and post-COVID-19 symptoms (cough, dyspnea, and fatigue), assessed at baseline and day 60. All patients were monitored for potential adverse effects associated with vitamin D supplementation.

### 4.7. Assessments (Primary, Secondary Outcomes)

All data were collected prospectively at the time of enrollment and throughout this study. Clinical and demographic information, including comorbidities, vaccination status, and symptoms associated with PCC was obtained directly from patient interviews and clinical evaluations. No retrospective data collection was performed.

Serum 25(OH)D levels were quantified by chemiluminescent microparticle immunoassay (CMIA) method using the Alinity 25-OH Vitamin D kit (Abbott Laboratories, Loughlinstown, Dublin, Ireland). Measurements were made at baseline and at day 60.

All patients underwent monthly clinical evaluations before starting supplementation and again on day 60, and received standard care throughout this study. In the daily dosing group, post-treatment blood samples were collected 24 h after the final dose of vitamin D supplementation. In the bolus dosing group, samples were collected within seven days of the last bolus dose. To ensure adherence, researchers made regular phone calls to participants to remind them to continue supplementation throughout the trial. To monitor the effect of cholecalciferol supplementation, serum samples were used to measure 25(OH)D concentrations at 30 and 60 days. The remaining serum and plasma samples were stored at −80 °C for the measurement of IL-6 levels and GPx activity. Baseline and final clinical assessments included anxiety and depression scores, evaluated using the Hospital Anxiety and Depression Scale (HADS).

#### 4.7.1. IL-6 ELISA Assay

Serum IL-6 levels were quantified using a human IL-6 ELISA kit according to the manufacturer’s protocol (SinoBiological, Wayne, PA, USA). Briefly, the reagents were equilibrated at room temperature (22–28 °C) before use. Each well was pre-washed with 300 µL of 1× Wash Buffer, followed by two additional washes. Standard dilutions (3.13–200 pg/mL) were prepared, and 100 µL of each standard or serum sample was added to the pre-coated wells. The plate was incubated for 2 h at room temperature. After three washes with Wash Buffer, 100 µL of secondary antibody was added, incubated for 1 h, and washed again. Next, 100 µL of Substrate Solution was added and incubated for 20 min in the dark. The reaction was stopped by adding 100 µL of Stop Solution, and the absorbance was measured at 450 nm. Serum IL-6 concentrations were expressed in pg/mL.

#### 4.7.2. GPx Activity Assay

GPx activity was measured using a glutathione peroxidase activity kit (Enzo Life Sciences^®^, Plymouth Meeting, PA, USA) according to the protocol described in [140]. Briefly, 20 µL of each sample, standard, or blank (diluent) was added to each well, followed by 20 µL of 10X× reaction mix, 140 µL of 1× assay buffer, and 20 µL of cumene hydroperoxide to initiate the reaction. Absorbance was measured at 340 nm at one-minute intervals over a 10 min period at room temperature. GPx activity was expressed in units per mL of plasma [140].

#### 4.7.3. Anxiety and Depression Assessment

The Hospital Anxiety and Depression Scale (HADS) was used to evaluate anxiety and depression. The scale consists of 14 items divided into two subscales: one for anxiety (HADS-A) and another for depression (HADS-D). Each subscale contains seven items, scored from 0 to 21. In this study, scores of 0–7 were classified as normal, 8–10 as borderline, and 11–21 as indicative of anxiety or depression, respectively [141].

### 4.8. Sample Size

Sample size estimation was based on the primary outcomes: GPx activity and serum IL-6 concentrations. A 95% confidence level, 5% margin of error, standard deviation of 0.5, and a minimum detectable difference of 0.11 were used to determine the required sample size. For a finite population of 88 long-COVID patients treated at the Laboratory of Translational Investigation in Aging and Fibrotic Diseases at INER during the study period, the estimated required sample size was 40 patients (20 per group). Of the 88 eligible patients, 54 agreed to participate and were randomized into this study.

### 4.9. Statistical Analysis

Data distribution was assessed using the Kolmogorov–Smirnov test for normality. Categorical variables were compared using Fisher’s exact test and Chi square, while continuous variables were analyzed using Student’s *t*-test or the Mann–Whitney U test. Paired *t*-tests were used for self-controlled analyses to assess within-group changes over time. A general linear model for repeated measures was applied to evaluate interactions and individual factor effects and Bonferroni test as post hoc analysis. We performed ANCOVA and covariable adjustment. Statistical analyses were performed using GraphPad Prism (v. 8.0.0, GraphPad Software, San Diego, CA, USA) and SPSS Software (v. 19.0, IBM Corp. Released 2015. IBM SPSS Statistics for Windows, Armonk, NY, USA). A *p*-value of ≤0.05 was considered statistically significant.

## 5. Conclusions

Vitamin D supplementation was associated with a significant increase in GPx activity in both dosing regimens, independent of changes in serum 25(OH)D levels. A decrease in IL-6 levels was observed in the bolus group, while daily vitamin D supplementation was associated with improvements in anxiety and depression scores. These findings suggest a possible association between vitamin D supplementation and reduced oxidative stress and inflammation in PCC patients. However, because this was a self-controlled study, further randomized controlled trials with larger sample sizes and multivariate analyses are needed to confirm these observations and determine the optimal supplementation strategy for managing PCC.

## Figures and Tables

**Table 1 ijms-26-04582-t001:** Clinical and demographic characteristics of the participants.

	Type of Supplementation		
	Bolus (n = 28)	Daily (n = 26)	*p* Value
Sex (male/female)	(20/8)	(12/14)	0.09
Age (years, min–max)	57 (40–75)	59 (43–74)	0.64
Smoking	32.1% (9)	19.2% (5)	0.35
Smoking index	1.5 ± 4.34	3.47 ± 8.99	0.31
Body mass index (BMI, kg/m^2^)	30.41 ± 4.71	30.11 ± 3.57	0.79
Obesity/Overweight	42.9% (26)	50% (26)	0.49
Diabetes mellitus type 2	28.6% (8)	30.8% (8)	1.0
Systemic arterial hypertension	28.6% (8)	34.6% (9)	0.77
Asthma	0	3.8% (1)	0.48
Chronic obstructive pulmonary disease	3.6% (1)	0	1.0
Number of COVID-19 infections per year			0.74
Not reported	25% (7)	34.6% (9)	
1	53.6% (15)	46.2% (12)	
2–3	21.4% (6)	19.2% (5)	
Number of COVID-19 vaccines received			0.23
Not reported	25% (7)	42.3% (11)	
2–3	57.1% (16)	34.6% (9)	
4–5	17.9% (5)	23.1% (6)	

**Table 2 ijms-26-04582-t002:** Serum vitamin D levels, cholecalciferol requirements, and daily doses for the treatment groups.

	Type of Supplementation		
	Bolus (n = 28)	Daily (n = 26)	*p* Value
	Mean ± SD	Mean ± SD	
Baseline vitamin D (25(OH)D, ng/mL)	19.20 ± 5.59	18.99 ± 5.04	0.88
Required increase in 25(OH)D levels (ng/mL)	30.94 ± 5.64	31 ± 5.04	0.87
Vitamin D (cholecalciferol daily dose IU)	6263.79 ± 1627	5796.36 ± 1437	0.27
Final vitamin D (25(OH)D, ng/mL)	42.64 ± 10.38	35.99 ± 8.85	0.024 *
Change in 25(OH)D levels (ng/mL)	23.99 ± 11.65	17.28 ± 9.28	0.037 *

* Statistical significance *p* ≤ 0.05; SD = standard deviation.

**Table 3 ijms-26-04582-t003:** Self-controlled analysis of the serum determination of GPx activity and IL-6 levels.

	All Patients		
	Pre-Treatment	Post-Treatment	*p* Value
GPx (U/mg protein)	182.42 ± 78.33	234.12 ± 113.76	0.0001 *
IL-6 (pg/mL)	4.92 ± 9.59	2.62 ± 6.4	0.07
	Bolus		
	Pre-treatment	Post-treatment	*p* Value
GPx (U/mg protein)	145.09 ± 40.74	205.07 ± 119.47	0.006 *
IL-6 (pg/mL)	6.97 ± 12.05	3.08 ± 6.44	0.01 *
	Daily		
	Pre-treatment	Post-treatment	
GPx (U/mg protein)	212.64 ± 88.77	257.64 ± 114.26	0.01 *
IL-6 (pg/mL)	3.02 ± 6.62	2.25 ± 6.37	0.30

* Statistical significance *p* ≤ 0.05; SD = standard deviation.

**Table 4 ijms-26-04582-t004:** Self-controlled analysis of anxiety and depression assessment using the Hospital Anxiety and Depression Scale.

	All Patients		
	Pre-Treatment	Post-Treatment	*p* Value
HADS Anxiety Score	4.04 ± 3.98	1.07 ± 2.12	0.0001 *
HADS Depression Score	3.81 ± 4.54	1.24 ± 2.35	0.001 *
	Bolus		
	Pre-treatment	Post-treatment	*p* Value
HADS Anxiety Score	3.25 ± 3.20	1.74 ± 2.64	0.09
HADS Depression Score	2.86 ± 3.30	1.68 ± 2.65	0.23
	Daily		
	Pre-treatment	Post-treatment	
HADS Anxiety Score	4.88 ± 4.59	0.35 ± 0.97	0.0001 *
HADS Depression Score	4.85 ± 5.46	0.77 ± 1.92	0.0001 *

* Statistical significance *p* ≤ 0.05. Values are expressed as the mean ± standard deviation.

**Table 5 ijms-26-04582-t005:** Post-COVID-19 symptoms before and after supplementation with vitamin D.

	All Patients		
	Pre-Treatment (n = 54)	Post-Treatment (n = 54)	*p* Value
Cough	31.48% (17)	22.22% (12)	0.27
Dyspnea	25.92% (14)	11.11% (6)	0.08
Fatigue	22.22% (12)	9.25% (5)	0.11
	Bolus		
	Pre-treatment (n = 28)	Post-treatment (n = 28)	*p* Value
Cough	32.1% (9)	21.4% (6)	0.36
Dyspnea	28.6% (8)	14.3% (4)	0.32
Fatigue	17.8% (5)	3.6% (1)	0.19
	Daily		
	Pre-treatment (n = 26)	Post-treatment (n = 26)	*p* Value
Cough	30.7% (8)	15.4% (4)	0.32
Dyspnea	23.0% (6)	7.7% (2)	0.24
Fatigue	26.9% (7)	15.4% (4)	0.49

## Data Availability

The original contributions presented in this study are included in this article. Further inquiries can be directed to the corresponding author.

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
