# Peer review of "Association of Vitamin D Supplementation with Glutathione Peroxidase (GPx) Activity, Interleukine-6 (IL-6) Levels, and Anxiety and Depression Scores in Patients with Post-COVID-19 Condition"

_ijms, 2025, doi:10.3390/ijms26104582_

Round 1
Reviewer 1 Report (New Reviewer)
Comments and Suggestions for Authors
This is a very important interventional clinical trial aimed at assessing the levels of antioxidant defense system markers (GPx) and inflammation markers (IL-6) and numerous PCC symptoms before and after vitamin D supplementation. The major drawback of the manuscript is weak statistical analysis. To address the hypothesis and some limitations of the study (lines 385-390), the manuscript requires more advanced statistical analyses. Please follow my comments:
- In lines 177 -179 the authors wrote "We hypothesize that vitamin D supplementation, independent of changes in serum 25(OH)D levels, is associated with modifications in GPx activity, IL-6 levels, and clinical symptoms in PCC patients." . The study included individuals with deficient and insufficient levels of vitamin 25(OH)D, and the patients' cholecalciferol dose was adjusted based on their baseline serum 25(OH)D concentration to achieve a target level of 50 ng/ml. 25(OH)D levels were monitored to avoid undesirable side effects accompanying cholecalciferol supplementation. Thus, to address this hypothesis results regarding 25(OH)D level at baseline and posttreatment should be presented. Moreover, additional analyses of correlations between baseline and posttreatment 25(OH)D and PCC symptoms, GPX activity and IL-6 level should be performed. And then, the authors will be able to accept or reject the hypothesis.
- The title of the manuscript is not informative. It should refer to major findings.
- The results described in lines 230-236 should be presented also in Table with p -values for particular symptoms.
- In Table 1 p values for "Number of COVID-19 infections per year" and "Number of COVID-19 vaccines received" are misleading. To which compared groups p=0.51 and 0.49 refer?
Author Response
This is a very important interventional clinical trial aimed at assessing the levels of antioxidant defense system markers (GPx) and inflammation markers (IL-6) and numerous PCC symptoms before and after vitamin D supplementation. The major drawback of the manuscript is weak statistical analysis. To address the hypothesis and some limitations of the study (lines 385-390), the manuscript requires more advanced statistical analyses.
Please follow my comments:
Comment:
In lines 177 -179 the authors wrote "We hypothesize that vitamin D supplementation, independent of changes in serum 25(OH)D levels, is associated with modifications in GPx activity, IL-6 levels, and clinical symptoms in PCC patients." . The study included individuals with deficient and insufficient levels of vitamin 25(OH)D, and the patients' cholecalciferol dose was adjusted based on their baseline serum 25(OH)D concentration to achieve a target level of 50 ng/ml. 25(OH)D levels were monitored to avoid undesirable side effects accompanying cholecalciferol supplementation. Thus, to address this hypothesis results regarding 25(OH)D level at baseline and posttreatment should be presented.
Response:
Dear reviewer:
We appreciate your valuable suggestion. As requested, we have added Table 2 to present the serum 25(OH)D levels both at baseline and after supplementation. This table also includes the percentage of participants who reached the target level of ≥50 ng/mL, providing clearer context regarding dose adequacy and treatment response.
Comment
Moreover, additional analyses of correlations between baseline and posttreatment 25(OH)D and PCC symptoms, GPX activity and IL-6 level should be performed. And then, the authors will be able to accept or reject the hypothesis
Response:
We appreciate your observation. As suggested, we conducted correlation analyses between serum 25(OH)D levels, both baseline and post-treatment, and PCC symptoms, GPx activity, and IL-6 levels. As shown in the attached results, no statistically significant correlations were found.
However, it is important to emphasize that our hypothesis was not based on a linear or proportional relationship between these variables. Instead, we aimed to assess whether vitamin D supplementation was associated with clinical and biochemical changes within subjects, in the context of a self-controlled design. The absence of correlation does not exclude a possible association, as the primary goal was to evaluate the overall clinical and physiological response following the intervention, rather than a direct dependence on specific serum concentrations.
Furthermore, the focus of our study lies in the clinical and biological relevance of the observed changes, which are consistent with previous findings on the modulatory role of vitamin D in inflammation, oxidative stress, and neuropsychiatric symptoms in post-COVID-19 patients. Despite the limitations of the study design, we believe our findings provide evidence aligned with the literature and highlight the potential of vitamin D as a useful and safe intervention in this clinical context.
Correlations between baseline 25(OH)D level and GPX activity, IL-6 level, Anxiety and depression (global) |
||||
|
Rho (p value) |
|
|
Rho (p value) |
IL-6 level |
0.07 (0.62) |
|
Anxiety |
-0.05 (0.71) |
GPX activity |
-0.10 (0.45) |
|
Depression |
0.001 (0.99) |
Correlations between post treatment 25(OH)D level and GPX activity, IL-6 level, Anxiety and depression (global) |
||||
|
Rho (p value) |
|
|
Rho (p value) |
IL-6 level |
0.04 (0.91) |
|
Anxiety |
-0.07 (0.76) |
GPX activity |
-0.15 (0.33) |
|
Depression |
0.37 (0.15) |
Comment:
The title of the manuscript is not informative. It should refer to major findings.
Response:
We appreciate the comment. The manuscript title has been revised to more clearly reflect the main findings of the study. It now highlights the observed association between vitamin D supplementation and changes in glutathione peroxidase (GPx) activity, interleukin-6 (IL-6) levels, as well as anxiety and depression scores in patients with post-COVID-19 condition.
Comment:
The results described in lines 230-236 should be presented also in Table with p -values for particular symptoms.
Response:
We appreciate your observation. As recommended, we have added Table 5, which presents the results previously described in lines 230–236. This table includes the corresponding p-values for each evaluated symptom to provide a clearer and more accurate presentation of the findings.
Comment:
In Table 1 p values for "Number of COVID-19 infections per year" and "Number of COVID-19 vaccines received" are misleading. To which compared groups p=0.51 and 0.49 refer?
Response:
We appreciate your observation. We have reviewed and corrected the data in Table 1, which led to adjustments in some p-values. Additionally, we improved the formatting for clearer presentation. Specifically, the p-values corresponding to the variables “Number of COVID-19 infections per year” and “Number of COVID-19 vaccines received” are derived from a 3×2 contingency table analysis. While frequencies for six individual cells are presented, the reported p-value reflects the overall result of the contingency analysis for each variable.
Reviewer 2 Report (New Reviewer)
Comments and Suggestions for Authors
This study showed that the effect of vitamin D supplementation on GPx activity, IL-6 levels and anxiety score in 54 patients with post-COVID-19 condition.
- The rationale for comparing the supplementation into bolus and daily regimens needs to be better explained (e.g. physiological mechanisms or metabolic differences). Moreover, the absence of a non-supplemented control group limits the ability to clearly determine the effect of the supplementation, although pre-post comparisons within groups are possible. Baseline vitamin D level seems to be considered before dividing supplementation groups into bolus and daily.
- Specific vitamin D concentration of both groups was not shown in this study. Although the proportion (%) of participants with vitamin D deficiency/insufficiency and range is presented by group, the specific pre- and post-supplementation circulating levels of vitamin D are not reported for each group. This information is essential for evaluating the effectiveness of the supplementation. It is recommended that the authors provide the mean and standard deviation of vitamin D concentrations before and after supplementation by group and assess whether there are statistically significant changes depending on the type of supplementation.
- The sample size of participants in this study is too small. And notable baseline differences of general characteristics and GPx activity raise concerns about internal validity. These limitations make it difficult to attribute the observed outcomes solely to the effect of vitamin D supplementation and hinder meaningful interpretation of the differences between the bolus and daily groups.
- Additionally, when evaluating the effect of vitamin D supplementation on GPx activity and anxiety score, potential confounding variables that may influence GPx activity and anxiety were not adjusted. Appropriate statistical adjustments should be considered for publication, and they are necessary to strengthen the validity of the conclusions.
- Overall, the presentation of the tables could be improved to enhance clarity and readability. There appear to be many horizontal and vertical lines in the tables, which may detract from visual clarity. Additionally, the placement of p-values could be improved (Table 2 ,3).
- Introduction part has too many paragraphs.
- In section 4.6 of the Methods, the formatting of the assessment subheadings is inconsistent: some subsections include specific subheadings while others do not.

Author Response
This study showed that the effect of vitamin D supplementation on GPx activity, IL-6 levels and anxiety score in 54 patients with post-COVID-19 condition.
Comment
- The rationale for comparing the supplementation into bolus and daily regimens needs to be better explained (e.g. physiological mechanisms or metabolic differences).
Response:
Thank you for this valuable observation. We have expanded the justification for comparing bolus and daily vitamin D supplementation regimens in the Introduction section of the revised manuscript. This includes a discussion of the pharmacokinetic and metabolic differences between the two strategies, which may influence the biological trajectory of vitamin D activity. Bolus administration typically leads to a more rapid and pronounced rise in serum 25(OH)D levels than daily dosing, potentially resulting in distinct biochemical and clinical responses. In the absence of a non-supplemented control group, this comparison provides a meaningful approach to evaluating different dosing strategies in patients with PCC.
Comment
Moreover, the absence of a non-supplemented control group limits the ability to clearly determine the effect of the supplementation, although pre-post comparisons within groups are possible.
Response
We agree that the absence of a non-supplemented control group limits our ability to definitively attribute observed effects to vitamin D supplementation. However, as stated in the Methods section, the Ethics Committee did not authorize a placebo group because all participants had confirmed vitamin D deficiency or insufficiency. Withholding treatment in such cases was deemed unethical. We have explicitly discussed this limitation in the revised Discussion section and clarified that, while pre-post comparisons provide useful insights, causal inference remains limited.
Comment
Baseline vitamin D level seems to be considered before dividing supplementation groups into bolus and daily
Response:
We appreciate the opportunity to clarify this point. While serum 25(OH)D levels were measured before randomization (to confirm deficiency and determine dose), they were not used as a stratification factor for assigning patients to the bolus or daily group. Randomization was performed independently using a computer-generated 1:1 allocation. Post-randomization, baseline 25(OH)D values were used solely to individualize the supplementation dose using a standardized formula. As shown in Table 2, baseline 25(OH)D concentrations were comparable between the two groups, confirming balanced allocation. This clarification has been added to the revised Methods section (4.2 and 4.3).
Comment
- Specific vitamin D concentration of both groups was not shown in this study. Although the proportion (%) of participants with vitamin D deficiency/insufficiency and range is presented by group, the specific pre- and post-supplementation circulating levels of vitamin D are not reported for each group. This information is essential for evaluating the effectiveness of the supplementation. It is recommended that the authors provide the mean and standard deviation of vitamin D concentrations before and after supplementation by group and assess whether there are statistically significant changes depending on the type of supplementation.
Response:
Thank you for your valuable comment. In response, we have added Table 2, which includes the mean and standard deviation of serum 25(OH)D concentrations before and after supplementation, separately for each group (bolus and daily). Additionally, we included p-values to assess statistical significance between pre- and post-treatment values. These changes aim to enhance transparency regarding the effectiveness of both supplementation strategies, as you suggested.
Comment
- The sample size of participants in this study is too small.
Response:
We acknowledge the limitations related to our sample size. However, power calculations indicated that our sample was sufficient to achieve at least 80% power for the primary outcomes (GPx activity and IL-6 levels) and secondary outcomes (anxiety and depression scores). These analyses support the reliability of our findings within the constraints of a pilot study.
Comment
And notable baseline differences of general characteristics and GPx activity raise concerns about internal validity. These limitations make it difficult to attribute the observed outcomes solely to the effect of vitamin D supplementation and hinder meaningful interpretation of the differences between the bolus and daily groups.
Response:
We appreciate your comment, as you suggested, we applied a covariate adjustment to evaluate the post-treatment comparative analysis of the type of supplementation (bolus vs. daily) for GPx activity, since only in this parameter were significant differences observed in the pre-treatment. Despite this adjustment, no differences were observed, as shown in Supplementary Table S1.
Comment:
- Additionally, when evaluating the effect of vitamin D supplementation on GPx activity and anxiety score, potential confounding variables that may influence GPx activity and anxiety were not adjusted. Appropriate statistical adjustments should be considered for publication, and they are necessary to strengthen the validity of the conclusions
Response
We agree that confounding variables such as age, BMI, sex, and comorbidities can influence the outcomes. However, our self-controlled design minimizes these effects, as comparisons are made within the same individual over a short period. Confounding variables that remain stable in that timeframe (e.g., age, sex) are inherently controlled. Additionally, we confirmed that primary and secondary outcomes met statistical power thresholds. For between-group comparisons, we applied ANCOVA with covariates (age and BMI) and adjusted GPx results using Bonferroni post hoc testing. Multivariate analysis was not conducted due to the small sample size, which limits its statistical appropriateness.
Comment:
- Overall, the presentation of the tables could be improved to enhance clarity and readability. There appear to be many horizontal and vertical lines in the tables, which may detract from visual clarity. Additionally, the placement of p-values could be improved (Table 2 ,3).
Response:
Thank you for your suggestion. We have revised the tables to enhance clarity by simplifying the visual layout—reducing the number of vertical and horizontal lines and improving the alignment of p-values. To avoid redundancy, we included the self-controlled analysis data in the main tables and presented the comparative analysis in the supplementary material.
Comment:
- Introduction part has too many paragraphs.
Response
Thank you for this editorial observation. We revised and condensed the Introduction to improve coherence and focus while retaining essential background information. The revised version now presents a clearer narrative of the rationale for our study.
Comment:
- In section 4.6 of the Methods, the formatting of the assessment subheadings is inconsistent: some subsections include specific subheadings while others do not.
Response:
We appreciate this feedback. The inconsistencies in subheadings have been corrected, and all subsections under section 4.7 now follow a uniform format to improve clarity and navigation.
Round 2
Reviewer 1 Report (New Reviewer)
Comments and Suggestions for Authors
Dear Authors,
You have addressed sufficiently to my comments.
In the Table 5, you have showed Post-COVID-19 symptoms before and after supplementation with vitamin D. As one can see, 9 participants had cough before treatment, and aftertreatment with vit D 6 participants had cough in the bolus group. You observed non significant reduction of caugh after vit D treatment in bolus. It is not clear whether all participants had PCCs, such as cough, dyspnea and fatigue, when look at the numbers in the brackets in Table 5, even if some participants had only one symptom. Please clarify.
Author Response
Comment
Dear Authors,
You have addressed sufficiently to my comments.
Response
Dear Reviewer, thank you for your comments to improve this manuscript.
Comment:
In the Table 5, you have showed Post-COVID-19 symptoms before and after supplementation with vitamin D. As one can see, 9 participants had cough before treatment, and after treatment with vit D 6 participants had cough in the bolus group. You observed non significant reduction of caugh after vit D treatment in bolus. It is not clear whether all participants had PCCs, such as cough, dyspnea and fatigue, when look at the numbers in the brackets in Table 5, even if some participants had only one symptom. Please clarify.
Response
We sincerely thank you for your comments, which allowed us to improve the clarity of the manuscript.
All participants included in this study had a confirmed diagnosis of post-COVID-19 condition (PCC) according to the definition proposed by the WHO, which establishes the presence of one or more persistent symptoms for at least 2 months after SARS-CoV-2 infection, with no alternative medical cause to explain them. These patients were recruited from the PCC follow-up program of the National Institute of Respiratory Diseases (INER), where they remained under clinical surveillance for 12 months after the acute phase due to the persistence of symptoms.
As described in the WHO consensus, PCC symptoms may fluctuate over time, appearing, disappearing and reappearing. For this reason, some patients did not have all the symptoms they had previously experienced at the start of the study. For this analysis, only the most common symptoms in the sample (cough, dyspnea and fatigue) were considered, which were evaluated only in patients who reported them at baseline, without taking into account the appearance of new cases during follow-up. This methodological choice was made with the aim of avoiding biases due to the temporal variability of the symptoms and preserving the validity of the analysis.
To clarify this information, we have modified the presentation of the data in Table 5 and improved the wording of its description, as well as the main text, to specify that the data in the pretreatment column correspond only to patients presenting each symptom at baseline. New changes to the manuscript are highlighted in yellow.
We hope that these clarifications and adjustments adequately address your comment, and we greatly appreciate your detailed and constructive review.

This manuscript is a resubmission of an earlier submission. The following is a list of the peer review reports and author responses from that submission.
Round 1
Reviewer 1 Report
Comments and Suggestions for Authors
Due to the great confusion actually existing in relation with the post-COVID syndrome, it is very important to consider manuscripts centered on this topic. On the other hand, I think that one way to decrease the existing confusion despite the high number of existing publications, is just to publish works with a very high quality and that clearly improve the actual knowledge. But it is not the case of the present manuscript, that in my opinion should be rejected for publication because its low quality.
The introduction is the part of a manuscript in which authors explain in a brief extension, what is the situation of the studied issue showing to the reader the importance of the presented study. As a general recommendation, the authors in the final part of the section have to expose their hypothesis and, in accordance with it, the main objective of the study. And, in the present manuscript it is not the case. In the introduction there is not a clear plot line, and it is basic and the main reason for this section in any manuscript.
The main weakness of the study is basically a methodological one. The authors are describing as a clear etiological relationship between normalization of vitamin D plasma levels and beneficial impact on clinical evolution of patients hospitalized COVID-19 that required ICU admission. However, they forget that the existence of an association does not directly imply an etiological relationship. The first reason why the authors cannot conclude a beneficial effect is that, according to the applied statistical analysis, they just apply an univariate analysis. Because of this reason, they have not excluded the interactions of potential confounding variables as the effect of ICU. The authors have to create a multivariate model to avoid confounding effect of not considered variables. It is also necessary to know the kind of study they have done: is it, as seems, a retrospective one?.
Post-COVID syndrome is an entity in a very imprecise way by different authors. All these authors agree that it is also an exclusion diagnosis. Because of this reason I do not understand the significance of spirometric parameters, and why are considered as outcomes for patients with post-COVID syndrome. The presence of spirometric alterations rule out the diagnosis of post-COVID syndrome. The author also should consider that the found differences are not clinically significant despite statistical significance and could be simply secondary to the usual general status recovery found after an ICU admission.
I think that, instead of analyzing the effect of vitamin D on the evolution of PCS, they are in fact focusing the work in the effect of vitamin D administration on patients with hospitalized COVID-19 that required ICU. In any case, if the intention is to evaluate the impact of a therapeutic strategy, the authors should compare two groups of patients with a similar clinical situation, one group with the therapeutic intervention and the other one without it. At least if they want to analyze the effect of vitamin D administration. Finally, and again from a conceptual point of view, the authors should explain why they administered vitamin D in an endovenous or oral way. Did they expect a different effect?. Why?. It should also be considered that the therapeutic strategy the authors are evaluating is not the administration of vitamin D, but the normalization of vitamin D plasmatic levels.
For me, the actual manuscript does not have a basic quality to be considered for publication.
Author Response
Response to the reviewer
Comment:
Due to the great confusion actually existing in relation with the post-COVID syndrome, it is very important to consider manuscripts centered on this topic. On the other hand, I think that one way to decrease the existing confusion despite the high number of existing publications, is just to publish works with a very high quality and that clearly improve the actual knowledge.
-But it is not the case of the present manuscript, that in my opinion should be rejected for publication because its low quality.
Response:
Dear reviewer
We appreciate your comments and understand the importance of publishing high quality studies that significantly contribute to the understanding of post-COVID-19 condition (PCC). Based on your observations, we have made substantial changes to the manuscript to improve its clarity and methodological rigor.
Regarding the quality of the study, we would like to emphasize that our research was designed as a prospective clinical trial, which presents additional methodological challenges, especially when studying an emerging condition such as PCC. Our study is registered with ClinicalTrials.gov, which strengthens its validity and methodological transparency. Furthermore, to our knowledge, this is one of the first studies to evaluate the effect of vitamin D supplementation on the modulation of pro-inflammatory cytokines and antioxidant enzymes in a Mexican population with PCC.
The study design included randomization and double-blinding to ensure the reliability of the results. However, due to restrictions imposed by the Ethics Committee of the institution, the inclusion of a placebo group was not allowed because most participants had vitamin D deficiency or insufficiency, and withholding a potentially beneficial treatment was considered unethical. This limitation is acknowledged in the manuscript and reflects the ethical challenges of conducting clinical trials in vulnerable populations.
Despite the lack of a placebo-controlled group, we used a self-controlled design, which has been used in other clinical trials to evaluate the effects of interventions in specific health conditions (Hayek et al., 2025; Chen et al., 2024; Eberhardy et al., 2024; Inci et al., 2023). Our results show significant differences in markers of oxidative stress and neuroinflammation, supporting the hypothesis that vitamin D supplementation may play a relevant role in regulating these processes in PCC patients. These findings are consistent with previous studies reporting beneficial effects of vitamin D in COVID-19 (Sarhan et al, 2022; Bahmani et al, 2022; Atanasovska et al, 2021).
We appreciate your feedback and have improved the wording of the manuscript to increase methodological clarity and emphasize the study's contribution to this emerging field. We hope that these changes allow for a reassessment of the quality and relevance of our work.
Comment
-The introduction is the part of a manuscript in which authors explain in a brief extension, what is the situation of the studied issue showing to the reader the importance of the presented study. As a general recommendation, the authors in the final part of the section have to expose their hypothesis and, in accordance with it, the main objective of the study. And, in the present manuscript it is not the case. In the introduction there is not a clear plot line, and it is basic and the main reason for this section in any manuscript.
Response
Dear Reviewer,
We sincerely appreciate your insightful comment on the structure and clarity of the Introduction. Based on your recommendation, we have thoroughly revised this section to ensure a more coherent and structured presentation of the topic under review.
In the revised version, we have clearly outlined the context and significance of PCC, emphasizing its public health implications and the need for therapeutic interventions. In addition, we have explicitly stated our research hypothesis and aligned it with the main objective of the study, ensuring a logical progression from background information to the rationale of the study.
We believe that these changes significantly improve the readability and scientific rigor of the Introduction. Thank you for your valuable feedback, which helped us to refine our manuscript.
Comment:
-The main weakness of the study is basically a methodological one. The authors are describing as a clear etiological relationship between normalization of vitamin D plasma levels and beneficial impact on clinical evolution of patients hospitalized COVID-19 that required ICU admission.
Response:
Dear Reviewer,
We sincerely appreciate your comment and fully recognize the importance of distinguishing between association and causality. We would like to clarify that our study does not establish an etiologic relationship between the normalization of plasma vitamin D levels and the clinical outcome of patients hospitalized in the ICU for COVID-19 or those with post-COVID-19 condition (PCC).
The primary objective of our study was to evaluate the effect of vitamin D supplementation on GPx activity, IL-6 levels, and clinical symptoms in PCC patients, comparing two dosing regimens (bolus and daily). We did not analyze the direct relationship between plasma vitamin D levels and clinical outcomes, as this was not part of our methodological design. The measurement of plasma vitamin D levels before and after treatment was only intended to confirm that most patients had low levels at baseline and to assess potential differences in the final levels achieved between the two supplementation regimens.
To avoid any misinterpretation, we have revised the wording of the manuscript to emphasize that our study focuses on evaluating the effects of vitamin D supplementation in PCC patients without attempting to establish a direct causal relationship with normalization of plasma vitamin D levels.
We greatly appreciate your comments, which have helped us to improve the clarity and precision of our work.
Comment:
-However, they forget that the existence of an association does not directly imply an etiological relationship. The first reason why the authors cannot conclude a beneficial effect is that, according to the applied statistical analysis, they just apply an univariate analysis. Because of this reason, they have not excluded the interactions of potential confounding variables as the effect of ICU. The authors have to create a multivariate model to avoid confounding effect of not considered variables.
Response:
Dear reviewer,
We sincerely appreciate your valuable comments and understand the importance of addressing potential confounding variables to ensure a robust statistical analysis.
The primary objective of our study was to evaluate the potential effect of vitamin D supplementation on plasma GPx activity and IL-6 levels in PCC patients and to compare bolus versus daily administration. Demonstrating an etiologic relationship or assessing the effect of ICU admission was not within the scope of our study. As a secondary objective, we aimed to explore the potential effects of vitamin D treatment on specific clinical variables, which we have now refined in this revised version based on your insightful feedback.
In terms of statistical analysis, in response to your comments, we used the general linear model for repeated measures, which allows for hypothesis testing on both between- and within-subject factors, while also allowing for the assessment of factor interactions and individual effects. Given the study design, our objectives, and the sample size, we believe this approach strengthens the validity of our findings.
As you rightly point out, this study has limitations, as do most clinical studies published in the literature. However, we observed interesting results that pave the way for further research and contribute to the repositioning of vitamin D in the management of PCC, particularly in our population. Importantly, vitamin D supplementation represents a potentially cost-effective and beneficial therapeutic alternative for these patients.
In addition, prior to conducting this study, our research group conducted a review on the benefits of vitamin D in COVID-19, focusing specifically on neuroinflammation, which is one of the proposed pathophysiological mechanisms associated with PCC (Quintero-Fabián S, et al., 2022).
We greatly appreciate your constructive feedback, which was instrumental in refining our study and improving its methodological rigor.
Comment:
-It is also necessary to know the kind of study they have done: is it, as seems, a retrospective one?.
Response:
Dear Reviewer,
We appreciate your comment and regret that the previous wording did not clearly convey the study design.
We confirm that our study was prospective. To ensure clarity, we have revised and improved the wording in the Methods section of the manuscript to explicitly describe the study design.
Your comments have been extremely helpful in improving the presentation of our work, and we sincerely appreciate your feedback.
Comment:
Post-COVID syndrome is an entity in a very imprecise way by different authors. All these authors
agree that it is also an exclusion diagnosis. Because of this reason I do not understand the significance of spirometric parameters, and why are considered as outcomes for patients with post-COVID syndrome. The presence of spirometric alterations rule out the diagnosis of post-COVID syndrome. The author also should consider that the found differences are not clinically significant despite statistical significance and could be simply secondary to the usual general status recovery found after an ICU admission.
Response:
Dear Reviewer,
We sincerely appreciate your valuable comment and recognize the complexity of assessing respiratory function in patients with PCC. You are correct in pointing out that the relationship between spirometric parameters and PCC remains poorly defined, with few studies addressing respiratory symptoms in this condition.
A study by Choi et al (2024) reported that reduced forced vital capacity (FVC) (<90%) and persistent dyspnea beyond 12 weeks were associated with the development of PCC; however, the observed association was weak. The authors suggested that this association may be due to subclinical pulmonary embolism, post-COVID pulmonary fibrosis, or emotional factors. Similarly, Behera et al (2024) found that post-COVID patients had respiratory symptoms with altered diffusing capacity for carbon monoxide (DLCO) and reductions in FEV1 and FVC, particularly in those with moderate to severe disease during the acute phase. However, these findings were also weak and likely confounded by pre-existing comorbidities.
In addition, we recognize that some of the differences seen in our study may be due to general recovery after ICU hospitalization rather than a direct effect of vitamin D supplementation. Furthermore, we did not observe significant improvements in dyspnea, fatigue, or cough after supplementation in our cohort.
Given your insightful observation and the lack of both statistical and clinical significance in our spirometric results, we have decided to remove these data from the manuscript to ensure that our conclusions remain focused and scientifically sound.
We sincerely appreciate your feedback, which was instrumental in refining our study and improving its clarity.
Comment:
-I think that, instead of analyzing the effect of vitamin D on the evolution of PCS, they are in fact focusing the work in the effect of vitamin D administration on patients with hospitalized COVID-19 that required ICU. In any case, if the intention is to evaluate the impact of a therapeutic strategy, the authors should compare two groups of patients with a similar clinical situation, one group with the therapeutic intervention and the other one without it. At least if they want to analyze the effect of vitamin D administration.
Response:
Dear Reviewer,
We appreciate your comments and understand your concerns about the focus of our study. Although the patients in our cohort had previously been in the ICU for a severe COVID-19 episode, recruitment occurred one year after hospital discharge when they continued to experience persistent symptoms consistent with PCC. These patients returned to the hospital primarily for anxiety and depression, symptoms that they did not have prior to their illness, but which emerged after their COVID-19 recovery. This justifies their inclusion in the PCC population rather than an acute COVID-19 cohort.
Regarding the comparison of a treated group with a non-treated control group, we emphasize that the Ethics Committee of the institution did not approve the inclusion of a non-treatment group, as it was considered unethical to leave PCC patients with persistent symptoms untreated, especially in a context where the majority of them had vitamin D deficiency or insufficiency. This was a limitation of the study, which was explicitly acknowledged in the manuscript.
Given this ethical constraint, we decided to perform a intra-subject comparison, evaluating the clinical status and biomarkers of patients before and after vitamin D supplementation. Despite this limitation, significant differences in oxidative stress parameters and neuropsychiatric symptoms were observed after the intervention, supporting the relevance of our findings. Furthermore, to the best of our knowledge, no previous clinical trials have investigated this approach in PCC patients, underscoring the importance of our findings. We sincerely appreciate your review and valuable suggestions to improve the clarity and rigor of our study.
Comment:
-Finally, and again from a conceptual point of view, the authors should explain why they administered vitamin D in an endovenous or oral way. Did they expect a different effect?. Why?.
Response:
Dear Reviewer,
We appreciate your comment and would like to clarify our decision to compare two different vitamin D administration regimens (bolus and daily dosing). Our interest in this comparison stemmed from the need to evaluate whether differences in vitamin D metabolism and catabolism depending on the administration regimen could influence clinical outcomes, given that previous studies have reported differential effects between these strategies.
For instance, a study suggested that daily vitamin D supplementation might be more effective in preventing COVID-19 than intermittent bolus administration (Griffin et al., 2021). Another study found that an initial bolus dose followed by daily supplementation was more efficient in increasing 25(OH)D levels compared to daily supplementation alone (Grégoire-Pelchat et al., 2021). Additionally, a clinical trial demonstrated that both bolus and daily administration increased 25(OH)D levels after six months of supplementation (Bruyère et al., 2015). Given this background, we aimed to determine whether these two regimens could generate different outcomes in patients with post-COVID condition (PCC).
With daily vitamin D administration, we expected better outcomes in neuropsychiatric conditions due to its sustained effect on serum 25(OH)D levels. Studies have shown that daily supplementation helps maintain more stable levels of this metabolite while preserving a more consistent vitamin D metabolite ratio (VMR: 24,25(OH)₂D₃/25(OH)D₃) over 28 days (Ketha et al., 2018). This ratio is considered a more accurate biomarker of vitamin D status than absolute 25(OH)D₃ levels. In our study, this strategy was associated with a significant reduction in neuropsychiatric symptoms (anxiety and depression) in PCC patients.
On the other hand, bolus administration has been observed to cause a rapid increase in the VMR, peaking within 7 to 28 days after administration, but without the same stability as daily supplementation. Additionally, 25(OH)D levels tend to decline more rapidly with this regimen (Ketha et al., 2018). In our study, bolus administration was associated with a significant reduction in IL-6, suggesting a possible differential impact on inflammation modulation.
Despite these differences, both treatment groups successfully corrected vitamin D deficiency, which may have contributed to the improvement observed in patients. These findings align with a recent clinical trial demonstrating that daily and biweekly vitamin D3 supplementation with similar cumulative doses was effective in treating symptomatic vitamin D deficiency in children (Mondal et al., 2024).
We greatly appreciate your observation, and we hope this clarification helps to better understand our methodological choice.
Comment
-It should also be considered that the therapeutic strategy the authors are evaluating is not the administration of vitamin D, but the normalization of vitamin D plasmatic levels.
Response:
Dear Reviewer,
We appreciate your observation and have carefully revised the manuscript to ensure that it is clear that the therapeutic strategy evaluated in this study was vitamin D supplementation and its impact on GPx, IL-6 levels, and clinical variables in patients with PCC.
While Table 2 presents plasma vitamin D levels before and after the intervention, this data was included only for sample characterization and not as a primary or secondary outcome. The comparison between administration regimens (bolus vs. daily dosing) was not intended to evaluate the normalization of vitamin D levels as a therapeutic success criterion but rather to explore whether the administration regimen influenced the effects on inflammatory biomarkers and oxidative stress markers.
To prevent any misinterpretation, we have adjusted the manuscript wording to reinforce that the study focused on vitamin D supplementation as a therapeutic intervention rather than on normalizing plasma vitamin D levels.
We greatly appreciate your comment, as it has helped us improve the clarity of our work.
Finally, we mention that all modifications made to the manuscript have been highlighted in green, including changes suggested by other reviewers and we also improved the overall wording.
References
Hayek G, Reglodi D, Goetz C, Perone JM, Csutak A. Ranibizumab treatment improves the reading speed of patients with neovascular age-related macular degeneration: A nonrandomized clinical trial using the Radner reading chart. J Fr Ophtalmol. 2025 Jan;48(1):104350. doi: 10.1016/j.jfo.2024.104350.
Chen G, Zhang W, Wu Q, Yu Q, Cai Y, Luo W, Xu J, Zhang L, Hong R. Application of citric acid can enhance the accuracy for 13C-urea breath tests in the diagnosis of Helicobacter pylori infection in Chinese patients. Sci Rep. 2024 Jun 25;14(1):14658. doi: 10.1038/s41598-024-64927-3.
Wang Q, Zhou Y, Jing F, Feng Y, Ma J, Xue P, Dong Z. Effects of acute-phase COVID-19-related indicators on pulmonary fibrosis and follow-up evaluation. Eur J Med Res. 2024 Dec 18;29(1):585. doi: 10.1186/s40001-024-02197-5.
Eberhardt TD, Lima SB, Pozzebon BR, Santos KPD, Silveira LBD, Soares RSÁ, Alves PJ. Heel skin microclimate control: Secondary analysis of a self-controlled randomized clinical trial. J Tissue Viability. 2024 May;33(2):305-311. doi: 10.1016/j.jtv.2024.03.006.
Inci H, İnci F. Effect of ozone therapy on neutrophil/lymphocyte, platelet/lymphocyte ratios, and disease activity in ankylosing spondylitis: a self-controlled randomized study. Med Gas Res. 2023 Apr-Jun;13(2):53-58. doi: 10.4103/2045-9912.344981.
Sarhan N, Abou Warda AE, Sarhan RM, Boshra MS, Mostafa-Hedeab G, ALruwaili BF, Ibrahim HSG, Schaalan MF, Fathy S. Evidence for the Efficacy of a High Dose of Vitamin D on the Hyperinflammation State in Moderate-to-Severe COVID-19 Patients: A Randomized Clinical Trial. Medicina (Kaunas). 2022 Sep 27;58(10):1358. doi: 10.3390/medicina58101358.
Bahmani E, Hoseini R, Amiri E. Home-based aerobic training and vitamin D improve neurotrophins and inflammatory biomarkers in MS patients. Mult Scler Relat Disord. 2022 Apr;60:103693. doi: 10.1016/j.msard.2022.103693.
Atanasovska E, Petrusevska M, Zendelovska D, Spasovska K, Stevanovikj M, Kasapinova K, Gjorgjievska K, Labachevski N. Vitamin D levels and oxidative stress markers in patients hospitalized with COVID-19. Redox Rep. 2021 Dec;26(1):184-189. doi: 10.1080/13510002.2021.1999126.
Quintero-Fabián S, Bandala C, Pichardo-Macías LA, Contreras-García IJ, Gómez-Manzo S, Hernández-Ochoa B, Martínez-Orozco JA, Ignacio-Mejía I, Cárdenas-Rodríguez N. Vitamin D and its Possible Relationship to Neuroprotection in COVID-19: Evidence in the Literature. Curr Top Med Chem. 2022;22(16):1346-1368. doi: 10.2174/1568026622666220401140737.
Choi MG, Joo YW, Kim MH, Park S, Shin YY, Chun EM. Prognostic Factors for Predicting Post-COVID-19 Condition in Patients With COVID-19 in an Outpatient Setting. J Korean Med Sci. 2024 Jan 15;39(2):e23. doi: 10.3346/jkms.2024.39.e23.
Behera N, Patra JK, Dash BK, Pattnaik M, Sahu D, Rambhoopal Reddy B. Clinico-radiological and pulmonary function assessment of post-COVID-19 patients with respiratory symptoms. J Family Med Prim Care. 2024 Aug;13(8):2912-2920. doi: 10.4103/jfmpc.jfmpc_1721_23.
Griffin G, Hewison M, Hopkin J, Kenny RA, Quinton R, Rhodes J, Subramanian S, Thickett D. Perspective: Vitamin D supplementation prevents rickets and acute respiratory infections when given as daily maintenance but not as intermittent bolus: implications for COVID-19. Clin Med (Lond). 2021 Mar;21(2):e144-e149. doi: 10.7861/clinmed.2021-0035.
Grégoire-Pelchat P, Pastore Y, Robitaille N, LeMay S, Khamessan A, Kleiber N, Nyalendo C, Gagné N, Alos N, Mailhot G. Comparison of two vitamin D supplementation strategies in children with sickle cell disease: a randomized controlled trial. Br J Haematol. 2021 Jan;192(2):385-394. doi: 10.1111/bjh.17119.
Bruyère O, Deroisy R, Dardenne N, Cavalier E, Coffiner M, Da Silva S, De Niet S, Reginster JY. A phase IV, two-armed, randomized, cross-over study comparing compliance with once-a-month administration of vitamin D3 to compliance with daily administration of a fixed-dose combination of vitamin D3 and calcium during two 6-month periods. Osteoporos Int. 2015 Dec;26(12):2863-8. doi: 10.1007/s00198-015-3205-z.
Ketha H, Thacher TD, Oberhelman SS, Fischer PR, Singh RJ, Kumar R. Comparison of the effect of daily versus bolus dose maternal vitamin D3 supplementation on the 24,25-dihydroxyvitamin D3 to 25-hydroxyvitamin D3 ratio. Bone. 2018 May;110:321-325. doi: 10.1016/j.bone.2018.02.024
Mondal KAP, Singh P, Singh R, Malhotra RK, Seth A. Daily versus fortnightly oral vitamin D3 in treatment of symptomatic vitamin D deficiency in children aged 1-10 years: An open labelled randomized controlled trial. Clin Endocrinol (Oxf). 2024 Nov;101(5):491-498. doi: 10.1111/cen.15124.
Reviewer 2 Report
Comments and Suggestions for Authors
The article is well written and interesting.
I only have a few comments:
A few lines on vitamin D metabolism and absorption could be added to the introduction. The authors choose to analyse 2 different types of administration that have different effects so adding a few sentences about how the use of different forms affects metabolism and end effects might be relevant. For example, here are two studies that analysed how different forms of vitamin D are absorbed: 10.3390/nutraceuticals3020023 and 10.3390/ijms24065322
Lines 80-82 are missing references, ‘Studies have shown’ but there is no reference to studies.
I suggest rewriting the discussion: lines 233-245 should be moved to the beginning; it is too long and with more focus on the results of other studies than on the results of this paper. I suggest shortening the discussion and give more attention and relevance to the results obtained in this study .
Author Response
Response to the Reviewer
Comment
The article is well written and interesting.
Response
Dear Reviewer,
Thank you for your kind comment. We appreciate your positive feedback and are pleased that you found the study well-written and interesting.
Comment
I only have a few comments:
A few lines on vitamin D metabolism and absorption could be added to the introduction. The authors choose to analyse 2 different types of administration that have different effects so adding a few sentences about how the use of different forms affects metabolism and end effects might be relevant. For example, here are two studies that analysed how different forms of vitamin D are absorbed: 10.3390/nutraceuticals3020023 and 10.3390/ijms24065322
Response
Dear Reviewer,
Thank you for your valuable comments. We have revised the Introduction to include a discussion of vitamin D metabolism and absorption, particularly in the context of different routes of administration. This addition helps to clarify the rationale for analyzing two different supplementation regimens and their potential impact on metabolism and clinical outcomes. In addition, we have included the references you kindly suggested to support this discussion. We appreciate your input, which helped improve the clarity and completeness of our manuscript.
Comment
Lines 80-82 are missing references, ‘Studies have shown’ but there is no reference to studies.
Response:
Dear Reviewer,
Thank you for your comment. We have revised the Introduction to improve clarity and added the appropriate references to support the statement in lines 80-82. We appreciate your feedback, which helped strengthen the scientific rigor of the manuscript..
Comment
I suggest rewriting the discussion: lines 233-245 should be moved to the beginning; it is too long and with more focus on the results of other studies than on the results of this paper. I suggest shortening the discussion and give more attention and relevance to the results obtained in this study.
Response:
Dear Reviewer,
Thanks for your insightful comment. We have restructured the discussion by moving lines 233-245 to the beginning, as suggested, to improve coherence and flow. In addition, we have shortened the discussion to reduce excessive focus on previous studies and to emphasize the relevance of our findings.
Finally, all changes made to the manuscript, including those suggested by other reviewers, are highlighted in green and we also improved the overall wording.